# Intolerance-of-uncertainty therapy versus metacognitive therapy for generalized anxiety disorder in primary health care: A randomized controlled pilot trial

**Sandra af Winklerfelt Hammarberg**[1,2]*, **Eva Toth-Pal**[1,2], **Markus Jansson-Fröjmark**[3], **Tobias Lundgren**[3], **Jeanette Westman**[1,2,4,5], **Benjamin Bohman**[3]

**1** Division of Family Medicine and Primary Care, Department of Neurobiology, Care Sciences and Society, Karolinska Institutet, Huddinge, Sweden, **2** Academic Primary Health Care Centre, Region Stockholm, Stockholm, Sweden, **3** Centre for Psychiatry Research, Department of Clinical Neuroscience, Karolinska Institutet, & Stockholm Health Care Services, Region Stockholm, Stockholm, Sweden, **4** Division of Nursing, Department of Neurobiology, Care Sciences and Society, Karolinska Institutet, Huddinge, Sweden, **5** The Department of Health Care Sciences, Marie Cederschiöld University, Stockholm, Sweden

* sandra.af.winklerfelt.hammarberg@ki.se

## Abstract

### Objective

This randomized controlled pilot study investigated the feasibility of a future full-scale RCT to compare the effects of intolerance-of-uncertainty therapy (IUT) and metacognitive therapy (MCT) in primary health care patients with generalized anxiety disorder (GAD). Preliminary treatment effects were also evaluated.

### Materials and methods

64 patients with GAD at a large primary health care center in Stockholm, Sweden, were randomized to IUT or MCT. Feasibility outcomes included participant recruitment and retention, willingness to receive psychological treatment, and therapists' competence in and adherence to treatment protocols. Self-reported scales were used to assess treatment outcomes, including worry, depression, functional impairment, and quality of life.

### Results

Recruitment was satisfactory, and dropout was low. On a scale from 0 to 6, participants were satisfied with participating in the study (M = 5.17, SD = 1.09). Following brief training, therapists' competence was rated as moderate, and adherence was rated as weak to moderate. From pre- to post-treatment, reductions on the primary treatment outcome measure of worry were of a large effect size and statistically significant in both the IUT and MCT conditions (Cohen's $d$ for IUT = -2.69, 95% confidence interval [-3.63, -1.76] and $d$ for MCT = -3.78 [-4.68, -2.90]). The between-group effect size from pre- to post-treatment was large and statistically significant ($d$ = -2.03 [-3.31, -0.75]), in favor of the MCT condition.

**Data Availability Statement:** Data cannot be shared publicly because of Swedish legal and ethical restrictions related to sensitive patient information. Data are available from Region Stockholm (contact via data protection officer Camilla Heise Löwgren, camilla.heise-lowgren@regionstockholm.se) for researchers who meet the criteria for access to confidential data.

**Funding:** This work was supported by NSV grants from Region Stockholm [numbers 20170770, 20180792]. The funding body played no role in the design of the study; the collection, analysis, and interpretation of data; or in writing the manuscript.

**Competing interests:** The authors have no competing interests to declare.

## Conclusion

It is feasible to carry out a full-scale RCT to compare the effects of IUT to MCT for patients with GAD in primary health care. Both protocols seem effective, and MCT seems superior to IUT, but a full-scale RCT is needed to confirm these conclusions.

## Trial registration

ClinicalTrials.gov (no. NCT03621371).

## Introduction

Across the world, primary health care is the setting where most people with common mental disorders receive diagnosis and treatment [1–3]. Generalized anxiety disorder (GAD) is one of the most common anxiety disorders in primary health care patients [4, 5]. Lifetime prevalence varies across the world, but approximately 5% of the population of European and North American countries meet the criteria for the disorder at some point during their lives [4, 6, 7]. GAD is characterized by excessive and uncontrollable worry in several domains of daily life, as well as other symptoms, including irritability, restlessness, difficulty concentrating, fatigue, muscle tension, and sleep disturbance [8, 9]. To receive a diagnosis of GAD, these symptoms should have been present most days and affected functioning for at least six months [9]. Persistent anxiety and inability to control worry are often associated with muscle tension, which can cause chronic pain and dizziness [10, 11]. GAD is also associated with other somatic health problems, including irritable bowel syndrome and medically unexplained symptoms [8, 11–13] Moreover, it can increase the risk of other mental health problems, such as sleep disorders, adjustment disorders, and depression [8, 14]. These health problems often motivate patients with GAD to frequently attend primary health care for somatic and psychiatric complaints [12, 15], which in turn may lead to extensive physical examinations and investigations [10, 11]. Previous research shows that GAD often goes undetected and thus untreated in primary health care patients [6, 8, 11, 12, 16]. Accordingly, the disorder leads to disability and suffering for these patients, as well as high costs for society [8, 17, 18].

Cognitive behavioral therapy (CBT) is an effective treatment for several anxiety disorders [19, 20]. However, in contrast to the guidelines for treating other anxiety disorders, guidelines for GAD recommend antidepressant medication as the first-hand treatment and CBT as a secondary choice [21, 22]. In primary health care, treatment for GAD thus consists mainly of prescribing antidepressants, such as selective serotonin reuptake inhibitors, and primary health care patients with GAD have limited access to CBT [6, 11].

Although CBT is not recommended as the first-hand treatment for GAD, previous meta-analyses suggest that it is an effective treatment [23, 24]. There are several CBT protocols for GAD. Each is based on a different theory about the factors most essential in developing and maintaining the disorder and thus focuses on different cognitive constructs and behaviors [14]. It is not clear which CBT protocol is most effective and most suitable in primary care settings. There are several systematic reviews of the different protocols [19, 23–26], but there are few studies per protocol, and many studies have small sample sizes and/or used waitlist controls. Moreover, to the best of our knowledge, no randomized controlled studies comparing different GAD treatment protocols have been conducted in primary health care settings [12].

One of the main CBT protocols used for GAD in Sweden is intolerance-of-uncertainty therapy (IUT) [27]. Studies show that IUT improves GAD symptoms significantly more than

waitlisting or applied relaxation [14, 28]. IUT is based on the theory that intolerance of uncertain situations is a main cause of GAD and perpetuates the disorder, and that people with GAD adopt worrying as a behavior to cope with these situations [29]. The protocol includes self-monitoring, evaluation of positive worry beliefs and avoidance behavior, exposure training and improved problem orientation. Patients learn how to separate their worries about uncertain situations into two categories: actual problems that are a part of life (e.g. meeting a deadline) and hypothetical situations (e.g. experiencing an accident). They learn to manage the actual problems with problem-solving techniques and receive imaginal exposure to the hypothetical situations [27, 29].

Another protocol for treating GAD is metacognitive therapy (MCT) [30]. At least two studies have found that in patients with GAD, MCT reduces worry better than other CBT protocols [31, 32]. MCT is based on the theory that two types of worry underlie GAD. Type 1 consists of worry in response to certain situations and is linked to positive metacognitive beliefs that worry is helpful in coping with the situation. Type 2 consists of worrying about worry ("meta-worry") and is linked to negative metacognitive beliefs that worry itself is uncontrollable and dangerous [30]. By challenging the positive metacognitions of type 1 worry and the negative metacognitions (e.g. by scheduling worry to experience if it is controllable) of type 2 worry, MCT aims to help patients find alternative coping strategies [33].

Although both IUT and MCT are well-established protocols, only one previous study has compared the effectiveness of IUT and MCT in the treatment of GAD [32]. Conducted in a psychiatric outpatient setting, that study found that MCT was superior to IUT in reducing worry and improving other outcomes [32].

A full-scale randomized controlled trial (RCT) designed to evaluate effects of CBT and MCT in primary health care patients with GAD could investigate whether CBT and/or MCT is as effective in this setting as in psychiatric care. People with GAD who seek primary care for their symptoms typically present with somatic complaints [10, 12] and thus may be less motivated to adhere to a CBT protocol than people with GAD who seek psychological treatment specifically for GAD in a psychiatric care setting. Moreover, such an RCT in primary health care that compares IUT and MCT has the potential to clarify which of the two protocols is more effective in treating primary health care patients with GAD. An RCT could also help point the way to modifications that could better tailor the treatment protocols for use in primary health care settings, such as delivering full treatment in fewer sessions, which might increase accessibility to and acceptability of CBT or MCT for this common mental health problem.

However, to the best of our knowledge, neither protocol has been evaluated in a primary health care setting. There are therefore several uncertainties about the feasibility of performing a full-scale RCT designed to compare the effectiveness of the two protocols for patients with GAD in primary health care, including recruiting and retaining participants [34]. Relatedly, we do not know whether primary health care patients with GAD are willing to receive psychological treatment when many of them seek care for somatic complaints in a setting where the primary focus is physical health. Moreover, there are uncertainties about therapists' competence in and adherence to CBT and MCT in this setting, as well as uncertainties about how much training they need to be able to provide adequate treatment according to the protocols, which is important for future implementation in primary care.

Thus, the primary aim of the present randomized controlled pilot trial was to investigate the feasibility of a future full-scale RCT designed to compare the effects of IUT and MCT in primary health care patients with GAD. Feasibility outcomes included participant recruitment and retention, willingness to receive psychological treatment, and therapists' competence in and adherence to treatment protocols. A secondary aim was to explore the preliminary effects

of the two treatments on measures of worry, depression, functional impairment, and quality of life.

## Materials and methods

### Setting

The study was conducted between 2018 and 2020 at Liljeholmen Primary Health Care Center, Stockholm, Sweden, which is one of the country's largest primary health care centers and serves 31,000 patients. The center has a team of therapists who assess and treat patients with mild to moderate common mental disorders in accordance with national clinical guidelines [21].

### Participants

Patients at Liljeholmen Primary Health Care Center with a primary diagnosis of GAD were invited to participate. Patients were excluded from the study if they were younger than 18 years; unable to speak Swedish; or had a severe psychiatric disorder (e.g., psychosis, bipolar disorder), cognitive impairment, substance use disorder, or other ongoing psychological treatment. Patients who had initiated or altered a psychopharmaceutical prescription less than six weeks prior to study inclusion were also excluded.

Because assessing treatment effectiveness was not a primary aim, a power calculation was not conducted, but a goal of recruiting at least 50 participants was set to enable us to assess feasibility and preliminary effectiveness.

### Treatments

Both treatments were protocol-based. The protocols (Table 1), were delivered as described in two publications [30, 35]. The therapists were given the flexibility of delivering the therapy in fewer sessions if their clinical judgment indicated that it was appropriate for the individual

Table 1. Overview of the structure and content of 10 sessions[a] of intolerance-of-uncertainty therapy and metacognitive therapy as delivered in the study.

| Intolerance-of-uncertainty therapy | | Metacognitive therapy | |
|---|---|---|---|
| Session no. | Content | Session no. | Content |
| 1 | Psychoeducation | 1–2 | Psychoeducation |
| | Worry awareness training | | Challenge uncontrollability beliefs |
| | | | Practice detached mindfulness |
| | | | Introduce worry postponement |
| 2–4 | Uncertainty recognition | 3–4 | Challenge uncontrollability beliefs |
| | Behavioral exposure | | Explore and ban maladaptive control/avoidance behaviors |
| | | | Challenge danger beliefs |
| 5–6 | Reevaluation of the usefulness of worry | 5–6 | Challenge danger beliefs |
| | | | Focus on reversing maladaptive strategies |
| 7 | Problem-solving training | 7–8 | Challenge positive beliefs |
| 8–9 | Imaginal exposure | 9 | Work on reversing residual symptoms |
| | | | Challenge positive beliefs |
| | | | Work on a new plan |
| 10 | Relapse prevention | 10 | Relapse prevention |

[a]Therapists were given the flexibility of delivering the therapy in fewer sessions or in up to a maximum of 12 sessions if their clinical judgment indicated that it was appropriate.

patient. The use of fewer sessions to administer the same content to primary care patients is consistent with the instructions in the original manuals [30, 35], and fewer sessions could potentially increase the feasibility of treatment delivery in primary health care. The protocol described in Dugas and Robichaud's book on CBT for GAD was used for IUT [35]. Wells's book on MCT, chapter 6 on GAD, was used for MCT [30]. Both IUT and MCT were provided individually for up to 12 sessions. Participants received treatment from a therapist other than the therapist who conducted the diagnostic assessment.

## Therapists

Therapists were recruited from the team that provides psychological treatment at Liljeholmen Primary Health Care Center. The mean age of the ten therapists was 43.9 years (SD = 7.9), nine were women, six were clinical psychologists, and four were social workers. The therapists had a mean of 10.1 years (SD = 5.8) of clinical experience, and 4.5 years (SD = 7.0) of experience working with patients with GAD. Four therapists self-selected to provide IUT and six self-selected to provide MCT. Therapists provided either IUT or MCT throughout the whole trial. All therapists had completed basic training in CBT, and one had completed advanced training to become a psychotherapist. Before recruitment and not as part of the present study, three therapists had participated in specific training in treating GAD: a one-day workshop on MCT (one IUT therapist), a one-day workshop on applied relaxation (one IUT therapist), and a two-day workshop on applied relaxation (one MCT therapist).

## Training and supervision

Although therapists already used the Mini International Neuropsychiatric Interview (M.I.N.I.) [36] as part of their ordinary diagnostic assessment procedure, prior to participant recruitment, therapists took part in a one-day workshop on the measure to ensure that they used it similarly. The workshop included didactic presentation and skills training. Prior to the start of the study, therapists also participated in a one-day workshop on the protocol that they would use (IUT or MCT). The workshops consisted of lectures and exercises and were conducted by two clinical psychologists. One had extensive experience of practicing IUT, and the other of practicing MCT. To facilitate protocol adherence, during the treatment period, therapists participated in 75-minute biweekly group supervision meetings that included feedback on audio-recorded treatment sessions. These meetings were held separately for the IUT and MCT therapists and were led by two clinical psychologists with previous training in the relevant treatment and at least three years of experience of providing CBT for patients with GAD.

## Assessments

**Feasibility.** Feasibility was assessed with several measures, including flow of recruitment, retention, participants willingness to receive psychological treatment, and therapist competence in and adherence to the treatment protocols. Participants completed an evaluation form post-treatment. It included items on perceptions of the number of self-report measures, the procedure of audio-recording the sessions, the pace of treatment, the extent to which participants believed that their problems with worry and anxiety had predominantly psychological causes, and overall satisfaction with treatment. Responses were made on a scale that ranged from 0 to 6, where 0 was the lowest and 6 the highest rating, and 3 represented "lagom," a Swedish term that generally has a positive connotation and means "just right" or "just enough." The evaluation form included a free-text item where participants could provide additional comments about the study or treatment.

All treatment sessions were audio-recorded so that therapists would not know which sessions would be selected to assess their competence in the treatment that they provided and their adherence to the protocol. For each therapist, three recordings of session five were randomly selected for assessment of both competence and adherence. Session five was chosen to ensure that a treatment session was selected. As competence and adherence are somewhat overlapping constructs [37], to optimize differential assessment, one assessor evaluated competence and another adherence in the same recorded session. This dual evaluation of the same session also helped control for session-specific effects.

Competence in IUT was assessed with the Cognitive Therapy Scale-Revised (CTS-R) [38], which includes 12 items and rates competence on a Likert scale that ranges from 0 (absence of feature, or highly inappropriate performance) to 6 (excellent performance, or very good even in the face of patient difficulties). Competence in MCT was assessed with the Metacognitive Therapy Competency Scale (MCT-CS) [39], which includes 18 items and rates competence on a Likert scale that ranges from 0 (not done/not assessable/not applicable) to 5 (very good level). Adherence to the treatment protocols was assessed with two measures that were developed for the present study, one for IUT and one for MCT. These measures were designed to cover the treatment content in sessions four to seven. This session frame ensured that the assessed session was a treatment session. Further, it allowed us to assess adherence to approximately the same content, although treatment could be conducted at different pace and consist of a different number of sessions, and the IUT protocol is structured in modules, whereas the MCT protocol is structured in sessions. Each measure included four items about 1) reviewing home assignments from the previous session, 2) discussing the treatment model or registration form, 3) practicing treatment interventions (e.g., behavioral exposure or cognitive reappraisal), and 4) agreeing on new home assignments for the next session. Responses were provided on the following scale: 0 (no adherence), 1 (low adherence), 2 (moderate adherence), 3 (high adherence), and 4 (very high adherence), with a description of each step of the scale. Assessment was conducted by four clinical psychologists and/or psychotherapists: one assessed competence in IUT; another, competence in MCT; a third, adherence to IUT; and a fourth, adherence to MCT. The assessors who coded competence had previous training in using the measures. The assessors who coded adherence to protocol had extensive experience of practicing the treatment that they assessed.

**Treatment effects.** Severity of worry was assessed with the 16-item Penn State Worry Questionnaire (PSWQ) [40]. The total score ranges from 16 (no pathological worry) to 80 (severe pathological worry). Severity of depressive symptoms was assessed with the 9-item Patient Health Questionnaire (PHQ-9) [41]. The total score ranges from 0 (no depressive symptoms) to 27 (severe depressive symptoms). Functional impairment was assessed with the 12-item WHO Disability Assessment Schedule (WHODAS) 2.0 [42]. The total score ranges from 0 (no impairment) to 48 (severe impairment in all daily activities). Quality of life was assessed with the 5-item Satisfaction with Life Scale (SWLS) [43]. The total score ranges from 5 (extremely dissatisfied) to 35 (extremely satisfied).

Severity of worry and depression was assessed pre-treatment, mid-treatment (following the fifth session), post-treatment, and at follow-up six months after the end of treatment. Functional impairment and quality of life were assessed pre-treatment, post-treatment, and at follow-up. At the follow-up assessment, participants completed a survey about any additional psychological, psychopharmaceutical, or other treatment for anxiety or depression that they had received following treatment completion. Data on medication and sick leave during the study period (from inclusion to follow-up) were collected from participants' medical records.

## Procedure

Recruitment started in spring 2018 and ended in autumn 2019. Treatments were completed in January 2020. The last 6-month follow-up, via regular mail and online forms, finished in August 2020.

Patients who visited their general practitioner at Liljeholmen Primary Health Care Center for mental health problems and/or medically unexplained symptoms were referred to the therapist team for diagnostic assessment with the M.I.N.I. [36] version 7.0, which is based on the Diagnostic and Statistical Manual of Mental Disorders, 5th edition (DSM-5) [9]. Patients received a primary diagnosis of GAD if they fulfilled the criteria for the disorder and their symptoms were not better explained by other comorbid or secondary mental or somatic conditions. Patients who fulfilled the criteria for GAD as their primary diagnosis were given oral and written information about the study, and those who met the study criteria were invited to participate. Patients who provided written informed consent were included and randomly allocated to IUT or MCT. For participant flow through the study, see Fig 1.

To allocate each participant, a research nurse phoned an independent assistant who had access to a randomization list generated by another independent assistant using an online service (www.sealedenvelope.com). Randomization was conducted with a 1:1 ratio, and the randomization list was created in blocks of four or six using a random order of block size. The result of the randomization (i.e., allocation to IUT or MCT) was put in sealed envelopes with a serial number on each envelope corresponding to the order in which participants were recruited. For each recruited participant, the assistant opened the envelope with the participant's corresponding serial number and allocated the participant to treatment. Both the research nurse and the assistant took notes on the participants' allocation to treatment. Following the completion of recruitment, the nurse's and the assistant's notes were compared to the randomization list and found identical.

At inclusion, the research nurse collected information in a survey format on the background characteristics of participants (Table 2). The research nurse also collected the pre-, post-, and follow-up assessments, including symptom severity scales and evaluation surveys. The mid-treatment symptom severity scales (PSWQ and PHQ-9) were collected by the participant's therapist.

## Statistical analyses

Statistical analyses were performed with SPSS (Version 27, SPSS Inc., Chicago, IL). Proportions, means, and standard deviations (SDs) were calculated for feasibility measures. Differences in session attendance between the IUT and MCT groups were investigated with an independent t-test and differences in dropout with a Fisher's exact test. A descriptive analysis investigating the normality of continuous outcome variables was performed. In the preliminary evaluation of treatment effects, multilevel modeling was used to estimate the effects of time and of time by group on continuous outcome measures from the pre-treatment to the post-treatment assessment and from the post-treatment to the follow-up assessment. The maximum likelihood method was used to estimate model parameters. A first-order autoregressive structure with homogenous variances provided the best fit and was thus used as the covariance structure. We started with a basic model with a fixed intercept, we then built models adding random intercept and slope, and finally an interaction term. Allowing intercepts to vary (i.e., random intercepts model) means that the outcome variable is at different levels prior to treatment (i.e., the symptom level varies across participants). Similarly, allowing slopes to vary means that the change trajectory may vary across participants. Finally, to investigate whether treatment condition moderated the effect, an interaction term of time (fixed effect; treatment

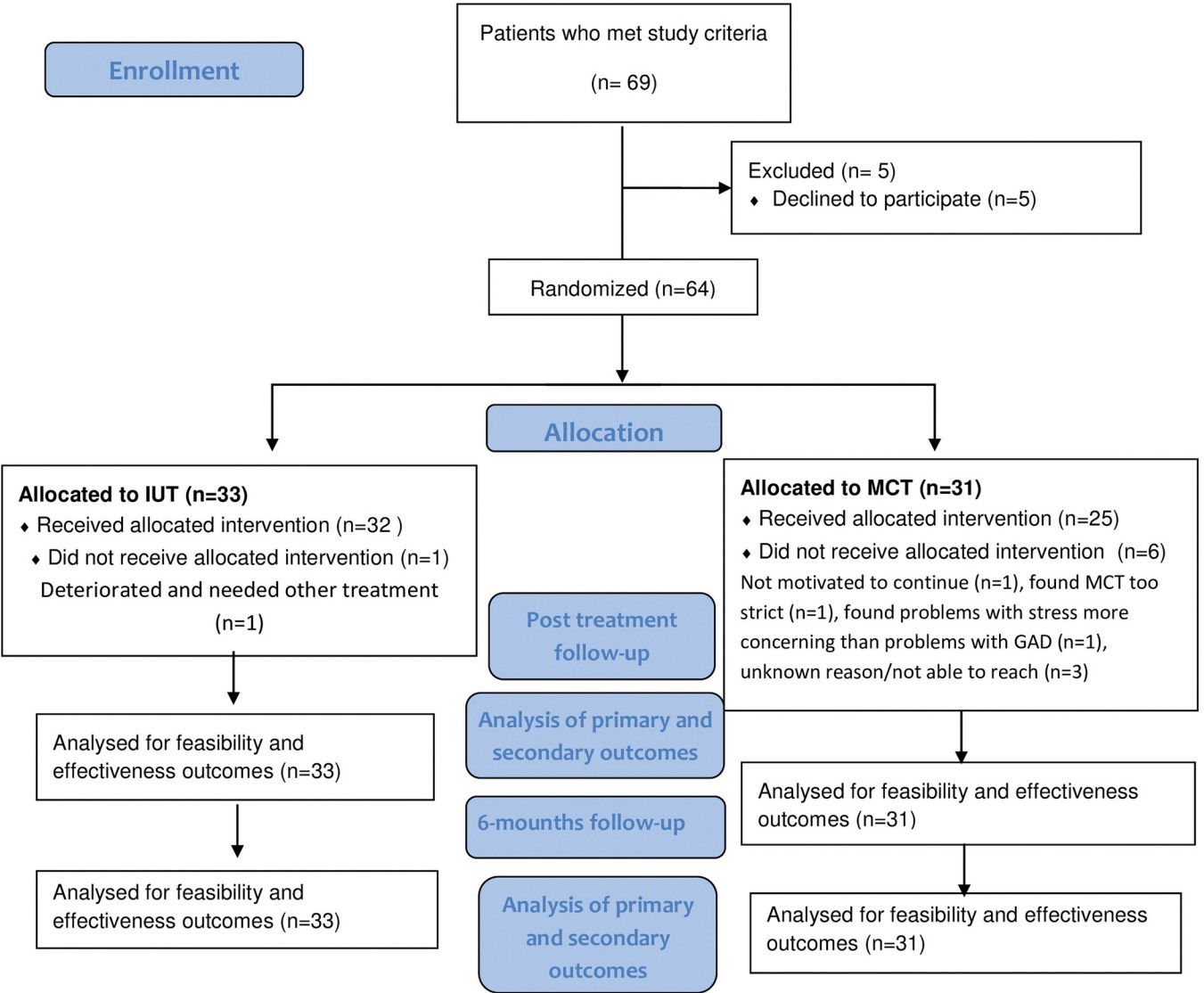

**Fig 1. Flow of participants in the study.**

period) and treatment condition (fixed effect; IUT or MCT) was added to the model with random intercept and slope. Each model's fit to observed data was evaluated with the likelihood ratio test. Models with significantly better fit than previous models were retained. Standardized effect sizes for between-group effects at mid-treatment, post-treatment, and follow-up were calculated as Cohen's *d* for multilevel models with the formula provided in Feingold [44], using the SD for the pooled sample at pre-treatment [45] and the pooled sample SD at post-treatment (for post-treatment to follow-up). For between-group effect size estimations, the beta coefficient (difference in change trajectories between treatments) was multiplied by the treatment duration at mid-treatment (5 weeks), the average treatment duration (9.6 weeks), or the follow-up duration (6 months), and then divided by the pooled SD of observed values at pre-treatment or post-treatment. For model-based *d*, 95% confidence intervals (CIs) were calculated with the formulas provided in Feingold [44]. In keeping with the principle of intention-to-treat, data from all participants were used in the multilevel models. Missing data were

**Table 2. Baseline characteristics of participants.**

| | IUT (n = 33) | MCT (n = 31) |
|---|---|---|
| Age [M (SD)] | 37.1 (11.6) | 35.7 (10.8) |
| Sex (female) | 81.8% | 80.6% |
| Civil status | | |
| • Married or cohabiting | 54.5% | 80.6% |
| • Living alone | 45.5% | 19.4% |
| Completed educational level | | |
| • High school or lower | 63.6% | 67.7% |
| • College or higher | 36.4% | 32.3% |
| Occupational status | | |
| • Employed or student | 90.9% | 90.3% |
| • Unemployed, on sick leave, or retired | 9.1% | 9.7% |
| Country of birth (Sweden) | 93.3% | 80.6% |
| Duration of GAD [years; M (SD)] | 14.8 (12.0) | 14.1 (13.2) |
| Previous CBT | 50.0% | 51.7% |

Abbreviations: IUT = intolerance-of-uncertainty therapy; MCT = metacognitive therapy; GAD = generalized anxiety disorder; CBT = cognitive behavioral therapy.

estimated using the maximum likelihood estimation, which is integrated in the multilevel modeling procedure in SPSS.

Treatment response was assessed with the reliable change index [46], which was calculated using the internal consistency [47] of the PSWQ and the sample SD of this measure at pre-treatment. A reliable change index (RCI) of $z \geq$ -1.96 indicates reliable improvement, whereas an RCI of $z \geq$ 1.96 indicates reliable deterioration. Differences in improvement and deterioration between groups at the post-treatment and follow-up assessment were investigated with Fisher's exact tests.

To assess recovery rates, two procedures were employed that combine statistically reliable change with clinically meaningful change. First, an RCI of 7 points and a cut-off of 53 points on the PSWQ as used in the study by van der Heiden and colleagues [32] were applied. Second, the same RCI and a cut-off of 47 points as used in the study by Nordahl and colleagues [31] (personal communication 21 March 2021) were applied. Differences in recovery rates between groups at post-treatment and follow-up assessment were investigated with Fisher's exact tests.

**Ethical approval and consent to participate.** Ethical approval for this study was obtained from the Regional Ethical Review Board in Stockholm, Sweden (2018/505-31). The research was conducted in accordance with the World Medical Association Declaration of Helsinki, and all patients and therapists provided written informed consent.

## Results

### Feasibility

Of the 69 patients who met the study criteria, 64 provided written informed consent and were included in the study. The flow of recruitment, an important measure of future RCT feasibility, was good, with an average of 4.6 (SD = 2.7, range = 1–10) patients consenting to participate and being included in the study per month from July 2018 to August 2019. The majority (81%) of participants were women, and the mean duration of GAD was 14 years (Table 2). Participants in the MCT group were to a higher degree married or cohabiting compared to the IUT group participants (Table 2).

Participants in the IUT group attended a mean of 10.5 (SD = 2.0) sessions, and participants in the MCT group attended a mean of 8.5 (SD = 3.1) sessions. An independent sample t-test

**Table 3. Estimated marginal means and standard deviations for the PSWQ, PHQ-9, WHODAS, and the SWLS from pretreatment to 6-month follow-up.**

|  | Group | Pre-treatment M (SD) | Mid-treatment (Session 5)M (SD) | Post-treatment M (SD) | Follow-up M (SD) |
|---|---|---|---|---|---|
| PSWQ | IUT | 67.9 (3.6) | 62.8 (5.3) | 55.8 (7.5) | 56.3 (10.3) |
|  | MCT | 64.9 (4.8) | 55.4 (5.8) | 42.1 (7.5) | 46.3 (8.3) |
| PHQ-9 | IUT | 9.5 (4.0) | 8.2 (3.6) | 6.4 (3.3) | 5.7 (3.2) |
|  | MCT | 10.3 (3.2) | 7.4 (2.8) | 3.3 (2.4) | 5.5 (2.7) |
| WHODAS | IUT | 23.2 (4.4) | Not assessed | 20.6 (3.4) | 18.9 (5.5) |
|  | MCT | 24.4 (4.2) | Not assessed | 14.9 (2.4) | 17.0 (3.9) |
| SWLS | IUT | 20.5 (3.7) | Not assessed | 22.1 (3.0) | 23.8 (3.7) |
|  | MCT | 19.2 (3.4) | Not assessed | 24.2 (2.5) | 25.2 (4.2) |

Abbreviations: PSWQ: Penn State Worry Questionnaire; PHQ-9: Patient Health Questionnaire 9; WHODAS: World Health Organization Disability Assessment Schedule 2.0; SWLS: Satisfaction with Life Scale; IUT: intolerance-of-uncertainty therapy; MCT: metacognitive therapy.

showed that participants in the IUT group attended significantly more sessions (t = 3.15, p = .003, d = 0.78). Seven participants (10.9%) dropped out during the treatment period (Fig 1). One participant in the IUT group dropped out after session 2. The remaining six dropped out from MCT, one after session 2, three after session 3, one after session 4, and one after session 10. The dropout rate was significantly higher in the MCT group ($\chi^2$ = 4.37, p = .037). Reasons for dropout are shown in Fig 1. Values were missing for 11 of the 14 variables (i.e., treatment outcome measures at different assessments). A total of 10.4% of values were missing, ranging from none (any outcome measure at pre-treatment) to 25.0% (SWLS scores at follow-up).

In the total sample, participants believed that the number of measures was "lagom" (just right) (M = 3.25, SD = 0.88). They viewed being recorded during the sessions neither positively nor negatively (M = 3.53, SD = 1.25) and thought the pace of treatment was "lagom" (M = 3.29, SD = 0.70). Furthermore, participants believed to some extent that their problems had predominantly psychological causes (M = 4.69, SD = 1.22). Overall, they were satisfied with participating in the study (M = 5.17, SD = 1.09). There were non-significant between-group differences in means (ts = -0.79–1.75, ps = .085-.735, ds = 0.09–0.46), except for the number of measures. Participants in the MCT group believed to a larger extent that there were too many measures than participants in the IUT group (*t* = -2.52, *p* = .015, *d* = 0.66). Optional free-text responses from both groups were mostly positive and indicated that participants were satisfied with the study and treatment and perceived the treatment as helpful. Eight of the 29 free-text responses were negative. Two of these negative responses were about difficulty filling out the questionnaires. The other six were about treatment-related challenges, including trouble understanding how to use specific techniques, feeling anxious when doing the exercises, and wanting more time and support to investigate their symptoms and beliefs in greater depth.

On average, the four IUT therapists treated eight participants each (range = 4–12), and the six MCT therapists five participants each (range = 2–10). For IUT therapists, the CTS-R item mean was 2.9 (SD = 0.4), representing a level of competence between "advanced beginner" and "competent", and close to the corresponding widely adopted competence threshold mean item score of 3 [37]. For MCT therapists, the MCT-CS item mean was 2.6 (SD = 0.4), representing a competence level between "weak" and "mediocre". However, standardized scores (z scores) for competence ratings showed no significant difference between the competence of IUT and MCT therapists (t = 0.88, p = .42, d = 0.10). For IUT therapists, the mean treatment adherence score was 2.7 (SD = 0.4), which represents moderate to high adherence. For MCT therapists, the mean adherence score was 1.9 (SD = 0.3), which is close to moderate adherence.

The adherence score was significantly lower for MCT than for IUT therapists (t = 2.56, p = .05, d = 2.29).

## Treatment effects

**Changes in symptoms of worry.** Within-group analyses showed that both IUT and MCT resulted in statistically significant reductions in the severity of worry as assessed with the PSWQ (Table 3).

We observed large within-group effect sizes from pre-treatment to mid-treatment (IUT: F (60.09) = 13.39, p = .001; MCT: F(55.46) = 21.83, p < .001) and from pre-treatment to post-treatment (IUT: F(58.82) = 32.54, p < .001; MCT: F(66.84) = 71.85, p < .001) (Table 4). Worry decreased more in the MCT group than in the IUT group (Table 4). A significant group by time interaction with a large effect size was observed from pre-treatment to post-treatment (F (128.92) = 9.88, p = .002), indicating that MCT was superior to IUT in reducing worry.

Between post-treatment and the 6-month follow-up, worry scores remained stable in the IUT group (F(57.81) = 0.01, p = .980) and the MCT group (F(46.39) = 1.18, p = .283) (Table 3). There was no significant between-group difference between these assessment points (F(109.98) = 0.79, p = .377) (Table 4).

At post-treatment, 71.9% of the participants in the IUT group and 96.3% in the MCT group met the criteria for reliable improvement (≥7 point decrease in PSWQ score). One participant in the IUT group and none in the MCT group met the criteria for reliable deterioration (≥7 point increase in PSWQ score). At the 6-month follow-up, 67.9% of the participants in the IUT group and 82.6% in the MCT group met the criteria for reliable improvement. At follow-up, one participant in each group met the criteria for reliable deterioration. Significantly more participants in the MCT group than in the IUT group met the criteria for reliable improvement at post-treatment ($\chi^2$ = 5.13, p = .031); the remaining reliable improvement and deterioration rates were not significantly different between the two groups ($\chi^2$ = 1.45, p = .336; $\chi^2$ = 0.86, p = 1.000; $\chi^2$ = 0.02, p = 1.000).

Regardless of whether we used a PSWQ cut-off score of 47 points, as in the Nordahl study [31], or 53 points, as in the van der Heiden study [32], more participants in the MCT group than the IUT group recovered. This was true both at post-treatment (47-point cut-off: MCT 59.3% vs. IUT 18.8%; 53-point cut-off: MCT 81.5% vs. IUT 37.5%), and at follow-up (47-point cut-off: MCT 60.9% vs. IUT 21.4%; 53-point cut-off: MCT 73.9% vs. IUT 39.3%). Irrespective

**Table 4. Within-group and between-group effect sizes with 95% confidence intervals for the PSWQ, PHQ-9, WHODAS, and the SWLS from pre-treatment to 6-month follow-up.**

| | Group | Within-group *d* (CI) Pre-treatment to mid-treatment (Session 5) | Within-group *d* (CI) Pre-treatment to post-treatment | Within-group *d* (CI) Post-treatment to follow-up | Between-group *d* (CI) Pre-treatment to mid-treatment | Between-group *d* (CI) Pre-treatment to post-treatment | Between-group *d* (CI) Post-treatment to follow-up |
|---|---|---|---|---|---|---|---|
| PSWQ | IUT | -1.78 (-2.75, -0.81) | -2.69 (-3.63, -1.76) | 0.03 (-0.49, 0.52) | -1.21 (-2.62, 0.19) | -2.03 (-3.31, -0.75) | 0.31 (-0.42, 1.07) |
| | MCT | -2.39 (-3.42, -1.36) | -3.78 (-4.68, -2.90) | 0.31 (-0.24, 0.83) | | | |
| PHQ-9 | IUT | -0.74 (-1.38, -0.10) | -0.62 (-1.15, -0.10) | -0.24 (-0.79, 0.32) | -0.38 (-1.35, 0.60) | -0.85 (-1.65, -0.08) | 0.73 (-0.18, 1.64) |
| | MCT | -1.34 (-2.09, -0.59) | -1.74 (-2.28, -1.20) | 0.54 (-0.11, 1.19) | | | |
| WHODAS | IUT | Not assessed | -0.48 (-0.96, 0.02) | -0.38 (-0.99, 0.23) | Not assessed | -1.27 (-1.96, -0.58) | 0.99 (0.09, 1.88) |
| | MCT | Not assessed | -1.81 (-2.31, -1.30) | 0.65 (-0.11, 1.30) | | | |
| SWLS | IUT | Not assessed | 0.36 (-0.21, 0.91) | 0.43 (-0.17, 0.95) | Not assessed | 0.76 (-0.11, 1.60) | -0.19 (-1.04, 0.76) |
| | MCT | Not assessed | 1.16 (0.51, 1.81) | 0.31 (-0.52, 1.04) | | | |

Abbreviations: PSWQ: Penn State Worry Questionnaire; PHQ-9: Patient Health Questionnaire 9; WHODAS: World Health Organization Disability Assessment Schedule 2.0; SWLS: Satisfaction with Life Scale; IUT: intolerance-of-uncertainty therapy; MCT: metacognitive therapy; d = model-based Cohen's d effect size

of which cut-off was used, between-group differences were statistically significant at post-treatment (47-point cut-off: $\chi^2 = 10.28$, p = .001; 53-point cut-off: $\chi^2 = 11.60$, p = .001) and at follow-up (47-point cut-off: $\chi^2 = 8.24$, p = .004; 53-point cut-off: $\chi^2 = 6.12$, p = .013). Thus, the results indicate that MCT reduced worry statistically and clinically significantly more than IUT.

**Changes in depressive symptoms.** Between pre-treatment and mid-treatment, IUT and MCT both statistically significantly reduced depressive symptoms as measured by the PHQ-9 (IUT: F(61.54) = 5.32, p = .024; MCT: F(57.00) = 12.97, p = .001) (Table 3). The same pattern was observed between pre-treatment and post-treatment (IUT: F(79.64) = 5.35, p = .023; MCT: F(81.64) = 39.32, p < .001). From pre-treatment to mid-treatment, there was no significant difference in change between groups (F(119.27) = 0.60, p = .441). However, from pre-treatment to post-treatment, MCT resulted in a larger reduction in depressive symptoms than IUT (F(158.16) = 4.73, p = .031). Effect sizes were moderate (IUT) to large (MCT) (Table 4). Between post-treatment and the 6-month follow-up, depressive symptoms remained stable in both groups (IUT: F(54.63) = 0.78, p = .382; MCT: F(47.18) = 2.61, p = .113), and there were no between-group differences in depressive symptoms during this time (F(108.78) = 2.66). Effect sizes were moderate.

**Changes in functional impairment.** Between pre-treatment and post-treatment, only MCT reduced functional impairment statistically significant as measured by the WHODAS (IUT: F(63.51) = 3.72, p = .058; MCT: F(58) = 51.29, p < .001) (Table 3). From pre-treatment to post-treatment, MCT resulted in a larger reduction in functional impairment than IUT (F(122) = 12.99, p < .001). The effect size was large (Table 4). Between post-treatment and the 6-month follow-up, functional impairment scores remained relatively stable in both groups (IUT: F(56.69) = 1.65, p = .204; MCT: F(46.15) = 4.00, p = .051). However, change in functional impairment scores in the IUT group indicated a slight improvement in function, whereas change in scores in the MCT group indicated a slight decline in function. The group by time interaction was significant (F(109.18) = 4.39, p = .039) and in favor of IUT. Effect sizes were large (Table 4).

**Changes in quality of life.** Between pre-treatment and post-treatment, MCT resulted in a significant improvement in quality of life as measured by the SWLS (F(55) = 13.01, p = .001), whereas IUT did not (F(63.84) = 1.58, p = .213) (Table 3). The effect size for MCT was large (Table 4). However, the between-group comparison for this period showed that MCT did not result in a larger improvement than IUT (F(119) = 3.04, p = .084) (Table 3). Between post-treatment and the 6-month follow-up, within-group quality-of-life scores remained stable (IUT: F(181.80) = 1.97, p = .162; MCT: F(45.56) = 0.55, p = .464). There were no between-group differences in quality of life during this period (F(331.14) = 0.12, p = .733).

**Changes in pharmaceutical treatment.** Consistent with the study protocol, pharmaceutical treatment for mental health problems remained stable during the treatment period for most participants. However, four participants in the IUT group initiated pharmaceutical treatment during the study of which three ceased pharmaceutical treatment prior to post-treatment. At post-treatment, 29 participants received pharmaceutical treatment for mental health problems. Nine of these 29 ceased pharmaceutical treatment by follow-up (three from the IUT group and six from the MCT group). Two participants in the IUT group reduced their dosage. One participant in the IUT group started pharmaceutical treatment at the follow-up assessment. Seven participants ceased all use of pharmaceuticals for mental health problems between post-treatment and follow-up; six of them were in the MCT group.

**Changes in sick leave.** At pre-treatment, six participants (two in the IUT group, four in the MCT group) were on sick leave between 25% of full time and full time: three for stress disorders, two for depression, and one for GAD. Of these participants, one in the IUT group

remained on full-time sick leave for depression at post-treatment and follow-up. In addition, one participant in the MCT group who had not been on sick leave at pre-treatment was on 50% sick leave for a stress disorder at post-treatment. At follow-up, one participant in the IUT group was on 50% sick leave for GAD, and one participant in the MCT group was on full-time sick leave for a stress disorder.

**Additional treatment received at follow-up.** At 6-month follow-up, four of the 48 participants who responded to the follow-up questionnaire (8.3%) had received additional CBT since the end of the study, two in each group. Twelve participants (25.0%) had received pharmaceutical treatment for anxiety or depression, seven in the IUT group and five in the MCT group. One participant had received other treatment for anxiety due to intimate partner violence.

## Discussion

### Main findings

The primary aim of the present pilot study was to investigate the feasibility of a future RCT designed to compare the effectiveness of IUT and MCT in primary health care patients with GAD. There were several uncertainties about feasibility because, to the best of our knowledge, neither treatment had previously been evaluated in a primary health care setting. These included participant recruitment and retention, willingness to receive psychological treatment, and therapists' competence in and adherence to treatment protocols. The results showed that patients with GAD were willing to participate. Recruitment was good, and the dropout rate was low. Participants reported that they were satisfied with taking part in the study and with the treatment they received. They believed to some extent that their symptoms had predominantly psychological causes, which suggests that it is feasible to provide psychological treatment to these patients in primary health care. Following brief training in the protocols, therapists in both groups showed some competence in the treatment they delivered. MCT therapists' adherence was rated significantly lower than that of IUT therapists.

The secondary aim of the present pilot study was to conduct a preliminary evaluation of the effects of the two treatments. Both IUT and MCT significantly reduced worry from pre-treatment to post-treatment, and the effect sizes were large. Similarly, depressive symptoms decreased significantly in both treatment groups; effect sizes were moderate for IUT and large for MCT. However, functional impairment declined significantly only in the MCT group. Similarly, quality of life increased significantly from pre-treatment to post-treatment in the MCT group only. From post-treatment to the 6-month follow-up, effects on outcome measures were maintained. MCT was the superior treatment for all outcomes from pre-treatment to post-treatment, and the between-group effect sizes were large, except for quality of life, which did not differ between treatment groups. From post-treatment to follow-up, the only significant difference between groups was in functional impairment, which improved in the IUT group. At post-treatment and follow-up, more MCT than IUT participants had a reliable reduction in worry and had recovered from worry. Moreover, not only was MCT the more effective treatment, but on average, it was two sessions shorter than IUT. If the RCT replicates these findings, it would mean that MCT is more effective than IUT and can be delivered in a shorter course of therapy, which would make it useful in primary health care.

### Comparison to other studies

**Feasibility.** The only previous study comparing IUT and MCT was conducted in a psychiatric outpatient setting and did not aim to test feasibility [32]. Dropout from MCT in the current study was similar to that observed in the van der Heiden trial [32]. Dropout from IUT in

the current study was substantially lower than in the van der Heiden study and also lower than the dropout rate in a meta-analysis of dropout in CBT [48]. The reasons the participants in the present study provided for dropping out do not explain why the dropout rate was lower in the IUT group. Nevertheless, the low dropout rate in the present study suggests that a future RCT in primary health care is feasible.

In the current study, following brief training in the protocols, the therapists from regular primary health care showed some competence in IUT and MCT, but the MCT therapists' mean adherence was assessed as significantly lower than that of the IUT therapists. This may represent a real difference or reflect the fact that the response scale of the adherence measures was not calibrated, and the assessors may thus have interpreted it differently. Furthermore, only recordings of session 5 were assessed. We will therefore take steps to improve inter-rater reliability and include treatment integrity checks throughout the treatment period in the future RCT. At least two independent assessors will assess each measure, and more than one session will be assessed. In the previous study [32], adherence but not competence was assessed; however, due to differences in assessment procedure and measures used, it is difficult to compare ratings.

**Treatment effects.** As in the current study, in earlier studies that compared MCT to CBT for GAD, all treatments reduced worry, but MCT reduced worry significantly more [31, 32]. Findings about reduced depressive symptoms and recovery from GAD were also similar to ours, as was the long-term maintenance of improvements [31, 32]. However, in the current study, the MCT group achieved this improvement in fewer sessions than the IUT group. The current study and the previous study that compared IUT and MCT both used a flexible number of treatment sessions: up to 12 in the present study and up to 14 in the earlier study [32]. Unlike our study, the previous study found little difference between the number of sessions of IUT and MCT attended by study completers. It is not clear why fewer sessions of MCT were clinically sufficient to achieve optimal improvement, but the explanation could be that in addition to being more effective in the short and long term, MCT might be more efficient, at least for some patients. However, a larger multicenter RCT is needed to confirm these findings, and the possible evidence that MCT is a useful treatment for patients with GAD in primary health care.

In this feasibility study, we did not assess potential mediators of effect, which could be of interest to include in the future RCT. However, one explanation for the better results in the MCT group may be that IUT only targets positive metacognitions about worry [35], whereas MCT targets both positive and negative metacognitions about worry, especially uncontrollability and danger beliefs [30, 33]. In the van der Heiden study, positive and negative metacognitions improved in both groups, but improved more in the MCT than the IUT group [32].

**Strengths and limitations.** This study had several strengths. Generalizability to other primary health care settings was strengthened by the recruitment of patients and therapists from regular primary health care and by keeping exclusion criteria to a minimum. In addition to measuring symptoms of worry and depression, the study also investigated functional impairment, quality of life, medication use, sick leave, and participants' perceptions of the treatments and of study participation. Moreover, several therapists were involved in treatment, all sessions were audio-recorded, the therapists were blinded to which session would be assessed, and the therapists' competence in and adherence to protocol were assessed by independent assessors.

The study also had several limitations. First, only one large primary health care center in an urban area participated, which reduces the generalizability of the findings to other primary health care populations. Moreover, despite randomization, the distribution is uneven between the groups regarding married/cohabiting versus living alone, which may have affected the

results. Second, therapists' competence in and adherence to the protocols was assessed in only a small proportion of the sessions as one of several feasibility measures, and inter-rater reliability was not ascertained. Third, due to the lack of an inactive control group, no conclusive causal inferences of the effects of IUT and MCT can be made; however, because of the differential effects it is likely that the treatments were at least partly responsible. Fourth, therapists were not randomized or allocated to perform both IUT and MCT, but rather chose which treatment they wanted to provide. Thus, their prior interest and competence in the treatment they selected could have biased the findings.

Finally, no sample size calculation was conducted, as the primary aim of the present study was to investigate feasibility. The mostly large effect sizes nevertheless suggest that the preliminary results regarding treatment effects are reliable. However, the lack of a sample size calculation constitutes a major limitation that may affect generalizability of the results.

To obtain a fuller picture of how treatment may improve the complex symptoms of GAD, in addition to assessing worry, future studies could investigate the severity of other common symptoms in patients with GAD, such as fatigue, sleep disturbance, irritability, and muscle tension. Because patients with GAD visit primary health care frequently for a variety of symptoms, future studies should also consider use of care as an outcome. Furthermore, future studies should consider using clinician-assessed measures rather than relying solely on self-report.

Even though pharmaceutical treatment was supposed to remain stable during the treatment period, some participants reduced or stopped medication, and others started medication, which could have affected the results. This finding underscores the need to monitor medication use during an RCT.

## Conclusions

We found that primary health care patients were willing to participate in a pilot study that compared two protocols for treating GAD. Therapists working in regular primary health care could provide protocol-based psychological treatment with some competence and adherence after brief training and regular supervision. We therefore conclude that it is feasible to carry out a full-scale RCT that compares the effectiveness of IUT and MCT for primary health care patients with GAD.

The preliminary evaluation of treatment effects suggests that both IUT and MCT effectively reduce worry, depression, and functional impairment, and increase quality of life for at least six months after treatment. However, MCT had superior effects on all outcomes, including recovery. MCT also achieved these outcomes in fewer sessions, which is relevant in primary health care setting. As these findings are similar to those of previous studies that compared MCT to IUT or other CBT in psychiatric outpatient clinics, we conclude that further studies of CBT, and specifically MCT, for primary health care patients with GAD are crucial to establish more knowledge and increase the availability of such treatments in primary health care.

## Supporting information

**S1 Checklist. Consort checklist.**
(DOC)

**S1 File. Original study protocol translated.**
(DOCX)

**S2 File. Original study protocol Swedish.**
(DOCX)

**S3 File. Translated ethical application.**
(DOCX)

## Acknowledgments

We thank our research nurse, Christina Stalby, and the participating patients and therapists at Liljeholmen Primary Health Care Center. We also thank scientific editor Kimberly Kane for useful comments on the text.

## Author Contributions

**Conceptualization:** Sandra af Winklerfelt Hammarberg, Eva Toth-Pal, Markus Jansson-Fröjmark, Tobias Lundgren, Jeanette Westman, Benjamin Bohman.

**Data curation:** Benjamin Bohman.

**Formal analysis:** Sandra af Winklerfelt Hammarberg, Eva Toth-Pal, Markus Jansson-Fröjmark, Benjamin Bohman.

**Funding acquisition:** Benjamin Bohman.

**Methodology:** Sandra af Winklerfelt Hammarberg, Eva Toth-Pal, Markus Jansson-Fröjmark, Tobias Lundgren, Jeanette Westman, Benjamin Bohman.

**Project administration:** Eva Toth-Pal, Benjamin Bohman.

**Resources:** Tobias Lundgren, Jeanette Westman.

**Supervision:** Jeanette Westman, Benjamin Bohman.

**Writing – original draft:** Sandra af Winklerfelt Hammarberg.

**Writing – review & editing:** Sandra af Winklerfelt Hammarberg, Eva Toth-Pal, Markus Jansson-Fröjmark, Tobias Lundgren, Jeanette Westman, Benjamin Bohman.

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
