## [Decision Letter · Decision Letter 0]

6 Feb 2023

PONE-D-22-31734Intolerance-of-uncertainty therapy versus metacognitive therapy for generalized anxiety disorder in primary health care: a randomized controlled pilot trialPLOS ONE

Dear Dr. af Winklerfelt Hammarberg,

Thank you for submitting your manuscript to PLOS ONE. After careful consideration, we feel that it has merit but does not fully meet PLOS ONE’s publication criteria as it currently stands. Therefore, we invite you to submit a revised version of the manuscript that addresses the points raised during the review process.

We look forward to receiving your revised manuscript.

Kind regards,

Walid Kamal Abdelbasset, Ph.D.

Academic Editor

PLOS ONE

Journal Requirements:

3. All Clinical Trials, Registered Reports, Registered Report Protocols, and Study Protocols carrying the CT flag must provide their original protocol as a Supporting Information file. Please upload a clean copy of the protocol with the confidentiality notice (and any copyrighted institutional logos or signatures) removed. PLOS Editorial Policy requires that the original protocol be published alongside your manuscript in the event of acceptance. Please note that should your paper be accepted, all content including the protocol will be published under the Creative Commons Attribution (CC BY) 4.0 license, which means that it will be freely available online, and any third party is permitted to access, download, copy, distribute, and use these materials in any way, even commercially, with proper attribution.

Reviewers' comments:

Reviewer's Responses to Questions

**Comments to the Author**

1. Is the manuscript technically sound, and do the data support the conclusions?

Reviewer #1: Partly

Reviewer #2: Yes

2. Has the statistical analysis been performed appropriately and rigorously? 

Reviewer #1: Yes

Reviewer #2: Yes

3. Have the authors made all data underlying the findings in their manuscript fully available?

Reviewer #1: Yes

Reviewer #2: Yes

4. Is the manuscript presented in an intelligible fashion and written in standard English?

Reviewer #1: Yes

Reviewer #2: Yes

5. Review Comments to the Author

Reviewer #1: The current paper describes a pilot RCT in comparing two interventions (IUT vs. MCT) based on CBT to treat GAD in the primary care setup. The paper is well written and follows the general guidelines of the pilot trials quite well. Some minor discrepancies were observed which is described in the review below.

1. Pilot RCT was not meant to evaluate either efficacy or effectiveness. Hence, correct the statement in line 162.

2. Both interventions are delivered via trained therapists. However, it is not clear how many therapists delivered the therapy for each arm. It is also not clear if a therapist is assigned to a specific subject throughout the study. This will have implications is the statistical analysis plan based on what is actually done and the therapist-level effect may need to be evaluated in the statistical model.

3. In a similar context it is not clear if a therapist is used to deliver both IUT and MCT to a different subject. Mixing up therapists may or may not affect contamination between the arms.

4. CONSORT diagram must also include all the data collection points including the primary endpoint and primary outcome(s).

5. It is reported that a multi-level model (also known as a mixed-effect model) is used. Please clearly state which is random and which one is a fixed effect (and why). Should your model also include the therapist-level effect?

6. Since the pilot study is not an efficacy and/or effectiveness-seeking study and not powered to detect any effect all the references of statistical significance, such as but not limited to p-value should be removed. The result of statistical details (line 351+) should be rewritten in light of the pilot/feasibility trial.

7. The authors may also give some specific details or key takeaways of the change/adaption that they recommend for conducting the larger powered trial from the lesson learned from the pilot/feasibility trial.

Reviewer #2: I would like to thank authors for their effort to conduct such a research work, however, I have some comments to be raised as follows:

- the authors did not mention how they calculate their sample size as even for the pilot studies there are rules for sample size calculation(1). please add it in details .

1- Whitehead AL, Julious SA, Cooper CL, Campbell MJ. Estimating the sample size for a pilot randomised trial to minimise the overall trial sample size for the external pilot and main trial for a continuous outcome variable. Stat Methods Med Res. 2016 Jun;25(3):1057-73. doi: 10.1177/0962280215588241. Epub 2015 Jun 19. PMID: 26092476; PMCID: PMC4876429.

- statistical analysis section:

* did the authors investigate the normality of data before proceeding in the inferential statistics used ??? as it was not mentioned in the original manuscript?!!

* authors did not mention the statistical strategies for managing missing data?? please add it to manuscript and explain how it had been managed

- Intervention:

-----------------

More details are needed even if the study followed previous published protocols ( references 30&35) to avoid distraction of the readers

6. PLOS authors have the option to publish the peer review history of their article (what does this mean?). If published, this will include your full peer review and any attached files.

Reviewer #1: No

Reviewer #2: No

---

## [Author Response · Author response to Decision Letter 0]

23 Mar 2023

In response to the journal requirements, we have adjusted the manuscript to meet the PLOS ONE’s style requirements, including those for file naming and figures. 

Due to Swedish legislation, the datasets generated and/or analysed during the current study are not publicly available, but are available on reasonable request from the corresponding author and/or the party responsible for the study, see ClinicalTrials.gov. Therefore, we have changed the data availability statement. 

According to PLOS ONE’s Editorial Policy, we have added the original and a translated version of the study protocol to the translated ethical application in the supporting information files.

Please find our point-by-point responses to the reviewers below. Changes in the manuscript are highlighted in grey.

Reviewers' comments:

Reviewer's Responses to Questions

Comments to the Author

1. Is the manuscript technically sound, and do the data support the conclusions?

Reviewer #1: Partly

Reviewer #2: Yes

2. Has the statistical analysis been performed appropriately and rigorously?

Reviewer #1: Yes

Reviewer #2: Yes

3. Have the authors made all data underlying the findings in their manuscript fully available?

Reviewer #1: Yes

Reviewer #2: Yes

Authors Response: Due to Swedish legislation, the datasets generated and/or analysed during the current study are not publicly available, but are available on reasonable request from the corresponding author and/or the party responsible for the study, see ClinicalTrials.gov. Therefore, we do not have the opportunity to provide files or links to these data and have changed the data availability statement. 

4. Is the manuscript presented in an intelligible fashion and written in standard English?

Reviewer #1: Yes

Reviewer #2: Yes

5. Review Comments to the Author

Reviewer #1: The current paper describes a pilot RCT in comparing two interventions (IUT vs. MCT) based on CBT to treat GAD in the primary care setup. The paper is well written and follows the general guidelines of the pilot trials quite well. Some minor discrepancies were observed which is described in the review below.

Response: Thank you.

Comment: 1. Pilot RCT was not meant to evaluate either efficacy or effectiveness. Hence, correct the statement in line 162.

Response: A secondary aim was to evaluate preliminary effectiveness, the sentence in line 162 has been revised accordingly. 

Comment: 2. Both interventions are delivered via trained therapists. However, it is not clear how many therapists delivered the therapy for each arm. It is also not clear if a therapist is assigned to a specific subject throughout the study. This will have implications is the statistical analysis plan based on what is actually done and the therapist-level effect may need to be evaluated in the statistical model.

Response: Participants were randomized to either IUT or MCT and received treatment from the same therapist throughout treatment, which was another therapist than the therapist performing the first assessment before randomization (lines 175 to 176.) However, therapists self-selected which treatment they would deliver. There were four therapists who self-selected to the IUT arm and six who self-selected to the MCT arm as described in line 186, a clarification has been added in line 187. We believe there is not sufficient power in our pilot trial to include a therapist effect in the models because of few therapists and few participants per therapist. 

Comment: 3. In a similar context it is not clear if a therapist is used to deliver both IUT and MCT to a different subject. Mixing up therapists may or may not affect contamination between the arms.

Response: We agree in the importance of these aspects and thus, no mixing of therapists was allowed in the study. We have tried to clarify this further in the methods section line 187: “Therapists provided either IUT or MCT throughout the whole trial”, and in the result section lines 400 to 401: “the four IUT therapists treated eight participants each (range=4-12), and the six MCT therapists five participants each (range=2-10)”. 

Comment: 4. CONSORT diagram must also include all the data collection points including the primary endpoint and primary outcome(s). 

Response: We have revised Fig 1, the CONSORT flow diagram and added the collection points including primary (feasibility) and secondary preliminary effectiveness outcomes. In keeping with the principle of intention-to-treat, data from all participants were used in the multilevel models. Missing data were estimated using the maximum likelihood estimation (lines 339 to 342)

Comment: 5. It is reported that a multi-level model (also known as a mixed-effect model). is used. Please clearly state which is random and which one is a fixed effect (and why). Should your model also include the therapist-level effect?

Response: We started with a basic model with a fixed intercept, then we built models adding random intercept and slope, and finally an interaction term. Allowing intercepts to vary (i.e., random intercepts model) means that the outcome variable is at different levels prior to treatment (i.e., the symptom level varies across participants); this is similar to including pre-treatment symptom level as a covariate in analysis of covariance. Similarly, allowing slopes to vary means that the change trajectory may vary across participants. Finally, to investigate whether treatment condition moderated the effect, an interaction term of time (fixed effect; treatment period) and treatment condition (fixed effect; IUT or MCT) was added to the model with random intercept and slope. This has been clarified on lines 320-327. We believe there is not sufficient power in our pilot trial to include a therapist effect in the models because of few therapists and few participants per therapist. 

Comment: 6. Since the pilot study is not an efficacy and/or effectiveness-seeking study and not powered to detect any effect all the references of statistical significance, such as but not limited to p-value should be removed. The result of statistical details (line 351+) should be rewritten in light of the pilot/feasibility trial.

Response: Because a secondary aim of the study was to explore preliminary effects, we would prefer to retain p-values and confidence intervals. Although the study was not powered to detect statistically significant effects, we believe that the interpretation of effects and effect sizes is facilitated by providing measures of statistical significance (i.e., p-values and confidence intervals). It is more the rule than the exception that pilot studies report statistical significance of effects. 

Comment: 7. The authors may also give some specific details or key takeaways of the change/adaption that they recommend for conducting the larger powered trial from the lesson learned from the pilot/feasibility trial.

Response: We agree in the importance of these aspects, which we discuss in separate parts of the discussion section. Most important in the future RCT is including more primary health care centers (line 586) and to do a power calculation for effectiveness measures (lines 618 to 619). Therapists’ competence could be tested before they are included in treatment, and therapists could be randomized rather than selecting protocol themselves (lines 615 to 618). Moreover, improvements of the inter-rater reliability in the assessments of therapist competence and adherence are needed. We intend to assess more sessions by at least two assessors for each measure as described in lines 566 to 570. In this feasibility study, we did not assess potential mediators of effect, which could be of interest to include in the future RCT (line 589 to 590). In lines 622 to 628 we discuss assessing further important outcomes for patients including somatic symptoms and use of care in a future study, and in lines 631 to 632 we underscore the importance of monitoring pharmaceutical treatment.

Reviewer #2: I would like to thank authors for their effort to conduct such a research work, however, I have some comments to be raised as follows:

Comment:- the authors did not mention how they calculate their sample size as even for the pilot studies there are rules for sample size calculation(1). please add it in details .

1- Whitehead AL, Julious SA, Cooper CL, Campbell MJ. Estimating the sample size for a pilot randomised trial to minimise the overall trial sample size for the external pilot and main trial for a continuous outcome variable. Stat Methods Med Res. 2016 Jun;25(3):1057-73. doi: 10.1177/0962280215588241. Epub 2015 Jun 19. PMID: 26092476; PMCID: PMC4876429.

Response: Thank you. The primary aim was to test feasibility, i.e. to test whether something can be done rather than to investigate the preliminary effects which typically can be the aim of a pilot [1]. As there was uncertainty about the feasibility of recruiting participants to an RCT in this setting, we set out to investigate the flow of recruitment as a feasibility measure. However, we set a goal of recruiting at least 50 participants to enable us to assess feasibility and preliminary effectiveness. In their paper, Whitehead and colleagues state that the “focus of this paper will be deriving pilot trial sample sizes based on a primary aim of the pilot being to estimate the standard deviation to be used for the main trial sample size calculation” (page 1058). This was not the primary aim of our pilot study; rather, it was to investigate the feasibility of a full-scale effectiveness trial in terms of participant recruitment and retention, willingness to receive psychological treatment, and therapists’ competence in and adherence to treatment protocols (lines 140 to 144). Thus, it seems that the paper of Whitehead and colleagues is not relevant in this context. Basing sample size on a primary aim of feasibility similar to ours is an acceptable procedure according to the CONSORT extension to randomized pilot and feasibility trials (Eldridge, et al., 2016). Moreover, there is controversy over the appropriateness of using data from pilot studies to inform sample size calculation in full-scale effectiveness trials ([2] e.g., Leon, Davis, & Kraemer, 2011). 

- Statistical analysis section:

Comment: -did the authors investigate the normality of data before proceeding in the inferential statistics used ??? as it was not mentioned in the original manuscript?!!

Response: Thank you for pointing that out. Yes, we investigated the normality of all outcome variables pre-treatment. A clarification has been added in line 313.

Comment: authors did not mention the statistical strategies for managing missing data?? please add it to manuscript and explain how it had been managed.

Response: In the manuscript, it is stated that “missing data were estimated using maximum likelihood estimation” (line 340). This estimation is automatic and integrated in the multilevel modeling procedure in SPSS. Additions have been made to clarify (lines 340 to 341). On lines 379 to 382, results are presented concerning missing values.

- Intervention:

-----------------

More details are needed even if the study followed previous published protocols ( references 30&35) to avoid distraction of the readers

Response: The contents of IUT and MCT are described in Table 1. The text in lines 97, 99 and 100 of the theory and methods of IUT has been clarified. An example has been added in lines 100-101 to clarify methods of MCT. 

6. PLOS authors have the option to publish the peer review history of their article (what does this mean?). If published, this will include your full peer review and any attached files.

Do you want your identity to be public for this peer review? For information about this choice, including consent withdrawal, please see our Privacy Policy.

Reviewer #1: No

Reviewer #2: No

References:________________________________________

 1. Eldridge SM, Lancaster GA, Campbell MJ, Thabane L, Hopewell S, Coleman CL, et al. Defining Feasibility and Pilot Studies in Preparation for Randomised Controlled Trials: Development of a Conceptual Framework. PloS one. 2016;11(3):e0150205.

2. Leon AC, Davis LL, Kraemer HC. The role and interpretation of pilot studies in clinical research. Journal of psychiatric research. 2011;45(5):626-9.

Sincerely,

Sandra af Winklerfelt Hammarberg on behalf of all the authors

---

## [Decision Letter · Decision Letter 1]

10 May 2023

PONE-D-22-31734R1Intolerance-of-uncertainty therapy versus metacognitive therapy for generalized anxiety disorder in primary health care: a randomized controlled pilot trialPLOS ONE

Dear Dr. af Winklerfelt Hammarberg,

Thank you for submitting your manuscript to PLOS ONE. After careful consideration, we feel that it has merit but does not fully meet PLOS ONE’s publication criteria as it currently stands. Therefore, we invite you to submit a revised version of the manuscript that addresses the points raised during the review process. Our original statistical reviewer was unavailable; thus, I reviewed your responses and revisions. I believed you adequately addressed this person's comments. Otherwise, our second reviewer had one final comment, which is worth addressing. Once you note that this study has limits related to generalizability (in the Limitations), I believe your manuscript will be ready for acceptance.

We look forward to receiving your revised manuscript.

Kind regards,

Ethan Moitra

Academic Editor

PLOS ONE

Journal Requirements:

Reviewers' comments:

Reviewer's Responses to Questions

**Comments to the Author**

1. If the authors have adequately addressed your comments raised in a previous round of review and you feel that this manuscript is now acceptable for publication, you may indicate that here to bypass the “Comments to the Author” section, enter your conflict of interest statement in the “Confidential to Editor” section, and submit your "Accept" recommendation.

Reviewer #2: All comments have been addressed

2. Is the manuscript technically sound, and do the data support the conclusions?

Reviewer #2: Partly

3. Has the statistical analysis been performed appropriately and rigorously? 

Reviewer #2: Yes

4. Have the authors made all data underlying the findings in their manuscript fully available?

Reviewer #2: Yes

5. Is the manuscript presented in an intelligible fashion and written in standard English?

Reviewer #2: Yes

6. Review Comments to the Author

Reviewer #2: thank you for your responding to the raised comments, however, there are some comments to be raised :

Regarding the sample size calculation; being the primary aim of the study is to investigate the feasibility, this does not give you an excuse for lacking sample size calculation. If the study was following CONSORT guidelines as described in manuscript and clarifies in supporting files (S1) , so, the sample size calculation should be addressed.

please address this as a major limitation in the limitation section as this will affect generalizability of the study.

7. PLOS authors have the option to publish the peer review history of their article (what does this mean?). If published, this will include your full peer review and any attached files.

Reviewer #2: No

---

## [Author Response · Author response to Decision Letter 1]

25 May 2023

Dear Associate professor Ethan Moitra and the Editorial Board,

Thank you for the opportunity to revise and resubmit the article, “Intolerance-of-uncertainty therapy versus metacognitive therapy for generalized anxiety disorder in primary health care: a randomized controlled pilot trial” (submission ID PONE-D-22-31734R1) to PLOS ONE. 

In response to the journal requirements, we have revised the manuscript to meet the comments of the reviewers. Please find our point-by-point responses to the reviewers below. Changes in the revised manuscript are highlighted in grey.

Sincerely,

Sandra af Winklerfelt Hammarberg on behalf of all the authors

Reviewers' comments:

Reviewer's Responses to Questions

Comments to the Author

1. If the authors have adequately addressed your comments raised in a previous round of review and you feel that this manuscript is now acceptable for publication, you may indicate that here to bypass the “Comments to the Author” section, enter your conflict of interest statement in the “Confidential to Editor” section, and submit your "Accept" recommendation.

Reviewer #2: All comments have been addressed

2. Is the manuscript technically sound, and do the data support the conclusions?

Reviewer #2: Partly

3. Has the statistical analysis been performed appropriately and rigorously?

Reviewer #2: Yes

4. Have the authors made all data underlying the findings in their manuscript fully available?

Reviewer #2: Yes

5. Is the manuscript presented in an intelligible fashion and written in standard English?

Reviewer #2: Yes

6. Review Comments to the Author

Please use the space provided to explain your answers to the questions above. You may also include additional comments for the author, including concerns about dual publication, research ethics, or publication ethics. 

Reviewer #2: thank you for your responding to the raised comments, however, there are some comments to be raised:

Regarding the sample size calculation; being the primary aim of the study is to investigate the feasibility, this does not give you an excuse for lacking sample size calculation. If the study was following CONSORT guidelines as described in manuscript and clarifies in supporting files (S1), so, the sample size calculation should be addressed. Please address this as a major limitation in the limitation section as this will affect generalizability of the study.

Response: Thank you for your valuable comments. We have revised the limitation section (lines 619 to 623 on page 28) to clarify this major limitation. The section now reads: “Finally, no sample size calculation was conducted, as the primary aim of the present study was to investigate feasibility. The mostly large effect sizes nevertheless suggest that the preliminary results regarding treatment effects are reliable. However, the lack of a sample size calculation constitutes a major limitation that may affect generalizability of the results.”

---

## [Editor Report · Decision Letter 2]

31 May 2023

Intolerance-of-uncertainty therapy versus metacognitive therapy for generalized anxiety disorder in primary health care: a randomized controlled pilot trial

PONE-D-22-31734R2

Dear Dr. af Winklerfelt Hammarberg,

We’re pleased to inform you that your manuscript has been judged scientifically suitable for publication and will be formally accepted for publication once it meets all outstanding technical requirements.

Kind regards,

Ethan Moitra

Academic Editor

PLOS ONE
---

## [Editor Report · Acceptance letter]

4 Jun 2023

PONE-D-22-31734R2 

Intolerance-of-uncertainty therapy versus metacognitive therapy for generalized anxiety disorder in primary health care: a randomized controlled pilot trial 

Dear Dr. af Winklerfelt Hammarberg:

I'm pleased to inform you that your manuscript has been deemed suitable for publication in PLOS ONE. Congratulations! Your manuscript is now with our production department. 

Kind regards, 

on behalf of

Dr. Ethan Moitra 

Academic Editor

PLOS ONE